# Internal and external validity of the brief version of the Multidimensional Personality Questionnaire: Exploratory structural equation modelling

**Rapson Gomez**[1], **Mark D. Griffiths**[2], **Vasileios Stavropoulos**[3] *

**1** Federation University Australia, Ballarat, Australia, **2** Nottingham Trent University, Nottingham, United Kingdom, **3** Victoria University, Melbourne, Australia

* Vasilisstavropoylos80@gmail.com

**Data Availability Statement:** All relevant data are within the manuscript and its Supporting Information files.

## Abstract

The present study used exploratory structural equation modelling (ESEM) to examine the theorized dimension structure of the brief version of the Multidimensional Personality Questionnaire (MPQ-BR) at the scale-level (i.e., 11 lower-order primary factors loading on four higher-order factors) and item-level (sets of 12 items loading on 11 lower-order primary factors). A total of 214 adults from the community addressed the MPQ-BR and the Behavioral Inhibition System (BIS)/Behavioral Approach System (BAS) scales. The findings revealed poor fit and poorly defined factors at the item-level alongside adequate fit and well-defined factors at the scale-level. The higher-order factors in the latter model were supported for external validity in terms of demonstrating the expected theoretical and empirical correlations with the scales of the BIS/BAS scales. Result related implications for professional application, as well as potential revisions of the MPQ-BF are illustrated.

One of the major models of personality is Tellegen's [1] multidimensional personality model [2]. Tellegen's model has 10 primary lower-order traits that map on to three higher-order (broad) personality dimensions, along with an eleventh primary trait that is considered as another higher-order dimension [2]. Viewed in this way, Tellegen's [1] model can be seen as having four broad higher-order dimensions. These dimensions are believed to be uncorrelated with each other (i.e., orthogonal). The higher-order and lower-order factors in Tellegen's [1] model are assessed using various versions of the Multidimensional Personality Questionnaire (MPQ; [1,3]), including a briefer (155-item) version of the MPQ (MPQ-BF; [4]). At the item-level, this measure has sets of 12 items loading on each of the 11 lower-order primary factors. At the scale-level, it has the 11 lower-order primary dimension loading on the four higher-order dimensions. To date, the proposed factor structures for this measure either at the item-level or the scale-level have not been clearly supported via exploratory factor analysis (EFA). Accordingly, the current research utilized a newly developed and more advanced technique called exploratory structural equation modeling (ESEM) to examine the factor structures of

**Funding:** The author(s) received no specific funding for this work.

**Competing interests:** The authors have declared that no competing interests exist.

the MPQ-BF at both item-level and scale-level. Contingent on at least adequate fit, the study also examined the external validity of the factors in the model/s.

## Tellegen's personality model

As aforementioned, Tellegen's model of personality has four higher-order or broad personality dimensions, and 11 lower-order primary trait factors. The higher-order personality dimensions are Positive Emotionality, Negative Emotionality, Constraint, and Absorption. Positive Emotionality involves dispositions towards positive emotions, and an appetitive approach. The lower-order sub-traits under Positive Emotionality include Wellbeing (inclination for positive emotionality/optimism), Achievement (working and enjoying hard and inclined to work, persistence, and perfectionism), Social Potency (inclination to be forceful, persuasive and influential, as well as aiming to be noticed by others), Social Closeness (being sociable, liking the company of others, being warm and affectionate, and seeking interpersonal comfort). Negative Emotionality refers to inclinations towards negative emotions, reactivity to stress and emotional liability, as well as defensive withdrawal. The primary lower-order trait factors for Negative Emotionality include Stress Reaction (e.g. tendencies for negative emotionality such as being nervous, worried, irritable, easily upset, moody, miserable, guilty and feeling worthless), Aggression (being physically aggressive and/or enjoying upsetting others), and Alienation (feeling mistreated-victimized, feeling prosecuted by others). Constraint refers to tendencies for behavioral restraint or low impulsivity. Its primary lower-order trait factors are Control (being reflective, rational, sensible, careful), Harm Avoidance (inclining to safe and familiar conditions and avoiding tensions), and Traditionalism (being conventional, displaying low rebellious nonconformity, and valuing good reputation). Absorption refers to emotional tendencies for innovative, imaginative and self-involving experiences and states, and it has a single primary trait factor, also called Absorption. It should also be noted that, together, the 11 primary traits address a diverse spectrum of dispositional features, behavior patterns, interpersonal behavior, and behavioral regulation [1,3], and the broad dimensions align with many psychobiological models of personality [2,4,5].

The MPQ-BF assesses Tellegen's model of personality and has 11 scales (with the same names) for each of the lower-order primary trait factors, alongside the four higher-order personality dimensions. Corresponding to Tellegen's model of personality, the MPQ allows for a comprehensive assessment of personality, covering traits related to temperament, interpersonal and imaginative style, and behavioral regulation. Also, the higher order dimensions maps onto constructs of emotion and temperament, which have direct reference to brain motive systems, and childhood models of temperament [6,7]. According to Patrick, Curtin and Tellegen [4], the MPQ has great potential to inform our understanding of the structure of personality, its genetic, neurobiological, and psychological underpinnings, and its relationship to different psychopathologies. To date, it has shown utility in increasing our understanding of how different personality dispositional are linked to different psychopathologies [7]. Given such qualities, it is critical that we have good understanding of the psychometric properties of the MPQ, especially its factor structure.

In the original scale introduction and validation research of the MPQ-BF [4], the dimensional structure at the lower-order primary trait scale-level of the MPQ–BF was examined in two samples using principal-components analysis (PCA). In both samples, and consistent with the theoretical proposed model, the primary trait scales loaded saliently on their target broad higher-order factors. However, the primary trait scales for Social Potency and Control also cross-loaded saliently and negatively on the higher-order dimensions of Constraint and Negative Emotionality, respectively. Absorption loaded on both the higher-order dimensions of

Positive Emotionality and Negative Emotionality. In a subsequent study, Eigenhuis, Kamphuis, and Noordhof [8] examined the factor structure of the primary trait scales of the MPQ–BF using EFA. As in Patrick et al.'s study, the primary trait scales loaded saliently on their target higher-order dimensions. However, there were considerable salient cross-loadings. Social Potency cross-loaded on the higher-order dimension of Constraint. Social Closeness loaded more strongly on the higher-order Negative Emotionality dimension, and Aggression had its primary loading on the higher-order Constraint dimension and not its designated higher-order Negative Emotionality dimension. Therefore, neither study showed clearly defined dimensions for the MPQ-BF at the primary trait scale-level. Also, as both these studies used PCA/EFA, the model fits were not available. Consequently, further investigation of the MPQ-BF at both the primary trait scale-level, and item-level are warranted.

## Exploratory, confirmatory and combined evaluation processes

EFA and CFA have traditionally been standard approaches for assessing the dimensional structure of an instrument. For a new measure, EFA (an exploratory approach) is generally used to ascertain its factor structure. Once this is reasonably and clearly defined, CFA (a confirmatory approach) is applied to confirm the hypothesized structure [9]. Methodologically, the EFA approach involves no limitation on cross-loadings of questions, and thus scale-questions are enabled to freely load across different dimensions. The standard CFA perspective for first-order factor models (usually referred to as the independent cluster model of CFA [ICM-CFA]) is a model-informed procedure. This approach allows research to assess for a priori described construct-conceptualization (generally suggested in the EFA). As such, items associate exclusively on their associated dimensions, and all the loadings on non-target dimensions (cross-loadings) are specified to zero [10]. The cross-loadings limitation in the ICM-CFA approach is deemed as a significant source of an analysis' compromise, as items are often not purely associated to their allocated latent dimensions, and thus varying levels of construct-close links with non-target but conceptually-similar dimensions can be envisaged [11]. Given that cross-loadings for MPQ-BF questions have been frequently identified in EFA evidence [4,6,7], it can be argued that the CFA approach would not allow the reality of ratings for the MPQ-BF to be expressed correctly. Consequently, it can be expected to support poor applicability, even when this may not be the case. Related to this, some researchers (i.e., [11,12]) have advocated that it is almost unrealistic to achieve acceptable fitting structures for sufficient multi-dimensional scored instruments, when assessed exclusively with ICM-CFA procedures. A more appropriate model would be one that allows for cross-loadings alongside testing the fit for the proposed/assumed structure. The ESEM approach was developed for this purpose [13].

ESEM is a combination of the EFA and CFA processes, merging the positives of the EFA (enabling cross-loadings) and the CFA (conceptualization-inspired) procedures. Available evidence has revealed the comparative analytical advantages of the ESEM process over the both the EFA and ICM-CFA procedures [12,14]. Therefore, it is conceivable that application of the ESEM approach is more likely to provide a more realistic and comprehensive evaluation of the structure of the MPQ-BF than PCA/EFA and CFA approaches. To date, only one study has examined the factor structure of the MPQ-BF using ESEM (i.e., [6]). At the item-level, this study showed that while CFA did not support the theorized 11-factor model, ESEM did provide support in terms of good global fit. However, as this study did not report the factor loadings for the ESEM solution, it is not known how well the factors were defined. This is critical because a model that is claimed to be supported needs to have clearly defined factors [14]. Additionally, because the study did not examine the factor structure of the MPQ-BF at the primary trait scale-level, there are no ESEM findings on the structure of the MPQ-BF at this level.

In addition to the factor structure, another significant measurement feature for a valid instrument is support for the external validity of the factors in the structural model. Generally, for this purpose, the external variables need to be linked conceptually or/and empirically with the factors in the model that is under investigation (in this case, the MPQ-BF). In this context, for the MPQ-BF factors, the personality constructs in Gray's [15] reinforcement sensitivity theory (RST) of personality [15,16] could be considered useful. In Gray's original RST theory, personality primarily refers to differences among individuals in two primarily underpinning neurobiological systems, namely the Behavioral Inhibition System (BIS) and the Behavioral Approach System (BAS). These systems have been linked with various types of reinforcements, emotions, behaviors, and personality. As originally conceived, the BIS is sensitive to signals of punishment, frustrative non-reward, and novelty. Furthermore, it underlies anxiety-related personality traits. The BAS is sensitive to signals of reward and non-punishment, and its arousal prompts approach behaviors toward these stimuli. It underlies personality traits related to impulsivity. Additionally, the BAS can be conceptually linked to Tellegen's Positive Emotionality factor [17]. While it may appear that the BIS is linked conceptually to Tellegen's Negative Emotionality factor, Tellegen [18] has instead likened the BIS to his dimension of Constraint. This is because this factor encompasses traits related to tendency towards restraint versus impulsiveness and venturesomeness. The constructs in Gray's original RST model of personality are generally assessed using the BIS/BAS scales [19]. In the BIS/BAS scales, the BIS and BAS scales assess their namesake constructs. The BAS scale has three subscales: BAS-Reward Responsiveness, BAS-Drive (BAS-DR) and Fun Seeking (BAS-FS). BAS-Reward Responsiveness assesses approach motivation in anticipation of a future reward; BAS-Drive assesses goal-directed behavior; and BAS-Fun Seeking assesses motivation to approach immediately (a form of impulsivity). Related to these associations, the study by Eigenhuis et al. [8] examined the relationships of the MPQ-BF broad factors with the factors in the BIS/BAS scales. Findings showed that Positive Emotionality was correlated positively with all three BAS scale scores; Negative Emotionality was correlated negatively with BAS-Drive and BAS-Reward Responsiveness scale scores, and positively with the BIS scale score; and Constraint was correlated negatively with the BAS-Fun Seeking scale score, and positively with BAS-Drive and BIS scale scores. Given these considerations, it can be speculated that Tellegen's Positive Emotional dimensions and its designated trait scales would be positively associated with BAS-Reward Responsiveness, BAS-Drive, and Bas-Fun Seeking. Also, Tellegen's Constraint dimension would be positively associated with the BIS, and negatively with BAS-Fun Seeking. Although Tellegen's Negative Emotional dimension is viewed as being theoretically associated with the BIS, Tellegen has not supported this association.

## The present study

Given the limitations in existing data concerning the MPQ-BF, the major aim of the present study was to use the ESEM procedure to examine the proposed factor structures at the item-level (i.e., items loading on the 11 lower-order primary trait factors) and the scale-level (i.e., the 11 primary trait scales loading on the four higher-order broad dimensions) among a large group of adults from the general community. In the ESEM model, as shown in Fig 1, the primary factor loads on their own targeted higher broad factors as well as non-targeted higher broad factors (at values close to zero). The present study examined the factors models separately at the item and scale levels because it is not possible to apply ESEM for second-order factor models using currently available SEM software programs, such as Mplus. Contingent on at least adequate model support, the external validity of the factors/dimensions in the model/s was also examined in terms of how the factors/dimensions were related to the scales in the

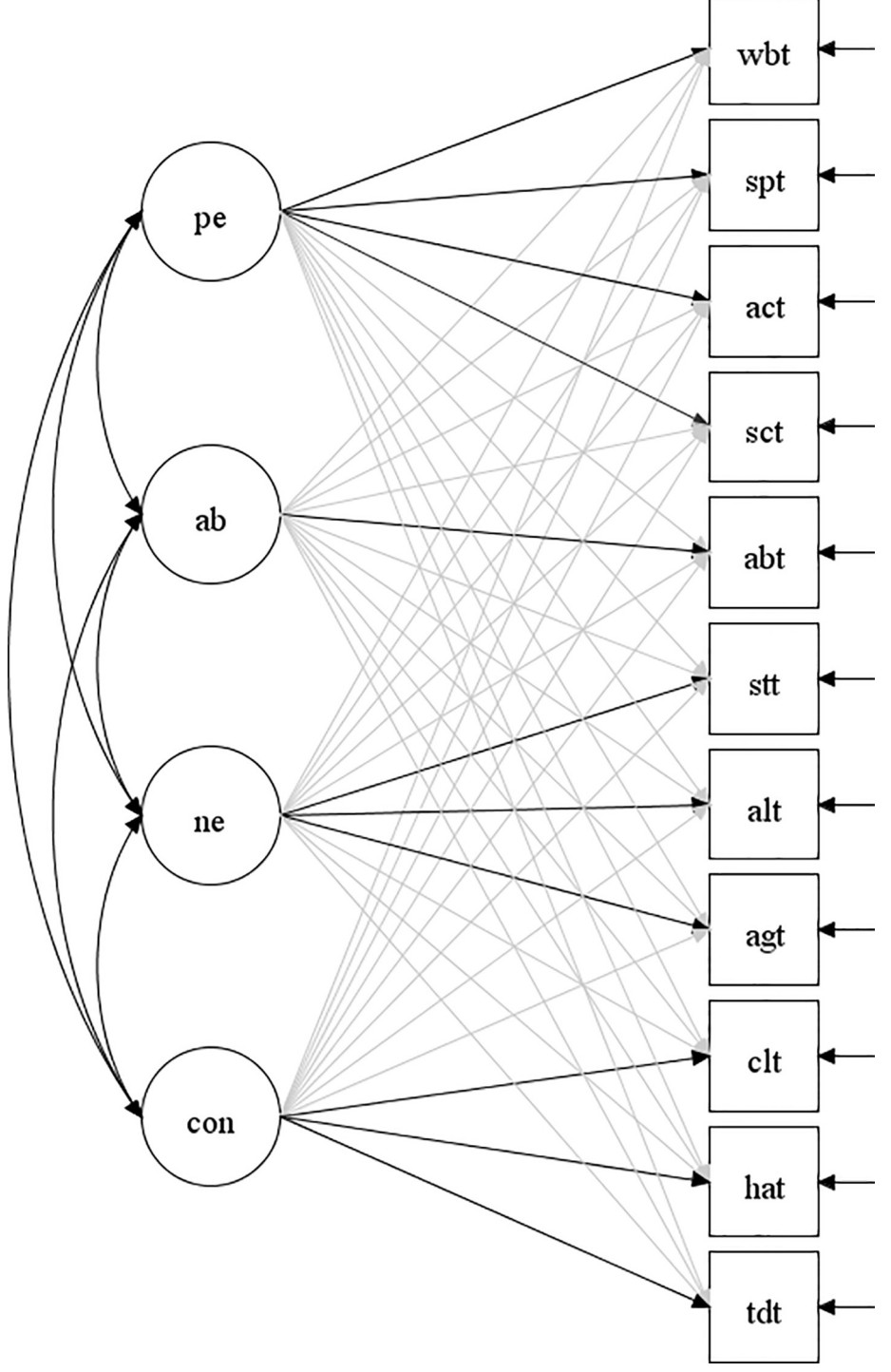

**Fig 1. Conceptual representation of ESEM model at the lower-order primary trait level tested in the study** (wbt = Wellbeing; spt = Social potency; act = Achievement; sct = Social closeness; abt = Absorption; stt = Stress reaction; alt = Alienation; agt = Aggression; clt = Control; hat = Harm avoidance; tdt = Traditionalism; pe = Positive Emotionality, ab = Absorption; ne = Negative Emotionality; con = Constraint. The darker arrows indicate the primary trait scale scores loading on their target higher-order factors, and the lighter arrows indicate the primary trait scale scores loading on non-target higher-order factors).

BIS/BAS scales. In terms of expectations, and based on the finding reported by Eigenhuis et al. [6], it was expected that ESEM would provide some level of support for the theorized MPQ-BF models at both the item-level and scale-level. Also, in terms of the external validity of the MPQ-BF factors, it was expected that the Positive Emotionality dimensions and its lower-order primary trait factors would be positively associated with the three BAS scales, and that the Constraint dimension and its lower-order primary trait factors would be positively associated with the BIS.

## Method

### Participants

Two-hundred and fourteen adults (115 females, $M_{age}$ = 34.64 years, $SD$ = 15.57; 99; males, $M_{age}$ = 36.91 years, $SD$ = 17.48; combined sample, $M_{age}$ = 31.71 years, $SD$ = 16.48, range = 18 to 76 years) were recruited in Australia through several sources from the State of Victoria (see Procedure). The 95% confidence interval maximum sampling error for a population of 214 is -+ 6.7% (Z = .196). The mean age of females and males did not vary significantly, $t$ (212) = 1.01, $p$ < .05. As shown in Table 1, for the sample as a whole, the mean scores for the 11 lower-order primary trait scales were all within the normal range.

### Materials

**Multidimensional Personality Questionnaire–brief (MPQ-BF; [4]).** As aforementioned, the MPQ-BF has 155 items and 11 primary personality trait dimensions. The personality trait scales and their allocation to the four higher dimensions (positive emotional temperament, absorption, negative emotional temperament, and constraint) were described above. It additionally embraces additional validity scale-items, assessing random responding (Variable Response Inconsistency), "yea-saying" or "nay-saying" (True Response Inconsistency), and social desirability (Unlikely Virtues). The $T$ scores corresponding to the total scores for each of the personality trait scales were exclusively used in the present study (as observed indicators in the ESEM model). Each MPQ-BF item was scored as either true (rated 1) or false (rated 0) regarding the applicability of the statement to the participant. Therefore, for every sub-scale

**Table 1. Standardized factor loadings in the ESEM Model.**

|  | Mean (SD) | Broad Factors | | | |
|---|---|---|---|---|---|
|  |  | **PE** | **AB** | **NE** | **CON** |
| Wellbeing | 49.59 (10.00) | **0.829** | 0.222 | -0.261 | 0.022 |
| Social potency | 50.33 (8.50) | **0.365** | -0.156 | 0.124 | -0.205 |
| Achievement | 49.79 (10.23) | **0.321** | 0.161 | 0.071 | 0.133 |
| Social closeness | 51.54 (10.13) | **0.267** | -0.238 | -0.196 | 0.131 |
| Absorption | 51.13 (9.06) | 0.157 | **0.702** | 0.152 | -0.127 |
| Stress reaction | 51.14 (10.39) | -0.286 | 0.283 | **0.486** | 0.130 |
| Alienation | 55.20 (9.06) | -0.174 | 0.159 | **0.750** | 0.053 |
| Aggression | 49.01 (9.720) | 0.112 | -0.257 | **0.608** | -0.164 |
| Control | 49.40 (9.79) | -0.146 | -0.034 | -0.107 | **0.503** |
| Harm avoidance | 48.20 (9.32) | -0.206 | -0.016 | -0.258 | **0.481** |
| Traditionalism | 45.75 (8.20) | 0.302 | -0.105 | 0.286 | **0.755** |

PE = Positive Emotionality, AB = Absorption; NE = Negative Emotionality; CON = Constraint. Bold values indicate targeted loadings. Underlined values indicated salient loadings (>.32).

and/or scale, increased ratings indicated increased presence of the specific traits. In the original scale development study, Patrick et al. [4] stated Cronbach's alpha values ranging from .75 to .84 for the MPQ-BF primary trait scales. In the present study, they ranged from .72 to .86.

**Behavioral Inhibition System/Behavioral Approach System (BIS/BAS) Scales.** The BIS/BAS Scales [19] examine individual variations considering the BIS and BAS traits of the original RST [15]. The BIS scale (seven questions) measures the inclination of experiencing negative affect and behavioral inhibition when indications for punishment or risk apply. The BAS scale (13 items) assesses BAS sensitivity (i.e., the inclination to exhibit strong positive affect and behavioral arousal in the context of reward expectations). The BAS scale involves three subscales: BAS-Reward Responsiveness (five items), BAS-Drive (four items) and BAS-Fun Seeking (four items). BAS-Reward Responsiveness assesses approach motivation in anticipation of a future reward; BAS-Drive assesses goal-directed behavior; and BAS-Fun Seeking assesses the tendency to impulsively pursue pleasure. Questions are scored on a four-point Likert-scale (1 = "very false for me" to 4 = "very true for me"). Higher subscales rates indicate higher sensitivities. The total sum scores for the BIS, BAS-Reward Responsiveness, BAS-Drive, and BAS-Fun Seeking were used in the present study. The BIS/BAS Scales possess satisfactory convergent, discriminant, and concurrent validity [19]. The internal consistencies (Cronbach's α) of the BIS, BAS-Reward Responsiveness, BAS-Drive, and BAS-Fun Seeking Scales in the present study were .74, .80, .84, and .78 respectively.

## Procedure

Following ethics approval from the Human Ethics Committee of the Federation (former Ballarat) University, participants were approached at various work and social situations. All participants were from various sources from the community. These included individuals enrolled at shopping centers, and sporting, recreational, hobby, and social cubs and associations. The procedure was explained and, if they were positive in participating, individuals were administered the questionnaires, the study plain language statement, and a prepaid reply envelope. The questionnaires included the MPQ-BF, and the BIS/BAS scales. Participants either mailed the questionnaires in prepaid envelopes or handed them to the researchers. A debriefing statement was distributed at the end of the study. Around 350 questionnaires were distributed to recruit the sample, resulting in a retention proportion approximating 61%.

## Statistical analysis

The ESEM and CFA models employed here were calculated with the M*plus* (Version 7) software [9], using target oblique rotation. Missing values were addressed based on the full-information maximum likelihood approach built into M*plus*. This method is based on that data is missing completely at random. As the scores at the item-level were categorical, weighted least square mean and variance adjusted chi-square (WLSMV) estimator was employed for the ESEM calculations at this level. WLSMV can address issues of lack of normality and is indicated for items with four or less response categories (binary items in this case; [9,20,21]). Because the scores for the primary traits were continuous, the robust maximum likelihood (MLR) estimator was employed for the analysis at the primary traits level. This robust estimator also corrects for potential lack of normality concerns.

To ascertain if there were positive indications for the ESEM model, the study examined global fit, salience of loading of the primary trait scales, and presence/absence of primary trait scales with salient cross-loadings. To be considered an acceptable model at item-level, it had to have at least acceptable global fit alongside most of the items having salient loadings on their respective targeted primary trait dimensions, and absence of items having secondary cross-

loadings on non-targeted primary trait dimensions. To be considered an acceptable model at the primary traits factor level, it had to have at least acceptable global fit, alongside most of the primary trait scale scores having salient loadings on their respective targeted broad factors, and absence of primary trait scale scores having secondary cross-loadings on non-target broad factors. Because all types of $\chi^2$ values, including $MLR\chi^2$, are distorted by large sample cohorts, the global fit of ESEM model was evaluated ultilizing the root mean square error of approximation (RMSEA) and the comparative fit index (CFI). Hu and Bentler [22] have advocated that RMSEA values approximating .06 or lower support good fit, close to .07 and up to < .08 moderate fit, close to .08 and up to .10 marginal fit, and close to or over .10 poor fit. For the CFI, rates of .95 or higher indicate good fit, rates between .90 and up to < .95 are indicative as acceptable fit, and rates < .90 are considered as indicating poor fit. For the present study, the CFI add RMSEA had to show at least acceptable and marginal fit, respectively, for it to be considered accepted. Tabachnick and Fidell [23] suggested that the rotated factor loading has to be at least .32 (approximately 10% of the overlapping variance between item/indicator and factor) to be meaningful. Therefore, factor loadings of .32 or above were considered salient in the study.

To test for the relationships of the factors in the ESEM MPQ-BF with the factors in the BIS/BAS scales, the correlations of these factors were computed. As is usual, positive correlations ($p <$ .05) for the ESEM MPQ-BF factors/dimensions with the factors in the BIS/BAS scales were taken an indicative of significant relationships. For these correlations, their effect sizes were interpreted in terms of Cohen's [24] guidelines for $r$ effect sizes: 0.1 = small, 0.3| = medium, and 0.5 = large.

## Results

### ESEM evaluation of the theorized model at the item-level

The fit values for the ESEM model were $\chi^2$ = 8715.51, $df$ = 7249, $p <$ 0.0001, CFI = 0.833, RMSEA = 0.022 (90% confidence interval = 0.020 to 0.024). For this model, the CFI value indicated poor fit, and the RMSEA value indicated good fit. Therefore, this model was interpreted as having inadequate fit. Table 2 shows the factor loadings for the MPQ-BF items in the ESEM model with 11 primary lower-order trait scales (factors). As demonstrated, not all the designated items loaded saliently on their respective primary trait factors. Additionally, all the primary trait factors also had several non-designated items loading saliently on them. Excluding non-salient items, negative salient items, and items that cross-loaded on other factors, there were (out of 12 in each case), 9, 6, 10, 7, 8, 6, 7, 6, 3, 6 and 7 salient items for the Wellbeing (Items 35, 50, 62, 74, 85, 97, 109, 121 and 144), Social potency (Items 2, 39, 61, 75, 87 and 110), Achievement (Items 3, 16, 27, 52, 76.88, 111, 123, 135 and 146), Social closeness (Items 28, 65, 89, 106, 124, 136 and 148), Stress reaction (Items 6, 18, 41, 78, 90, 101, 113 and 125), Alienation (Items 36, 42, 54, 91, 126 and 15), Aggression (Items 8, 20, 31, 67, 115, 127 and 139), Control (Items 9, 44, 56, 118, 128 and 140), Harm avoidance (Items 46, 57 and 141), Traditionalism (Items 23, 35, 58, 82, 94 and 154), and Absorption (Items 13, 24, 71, 83, 95, 107 and 119) factors, respectively. Consequently, all factors were not well defined. Given this, and that at the global level, the ESEM model did not show acceptable fit. Therefore, these findings showed lack support for the ESEM model at the item-level.

### CFA and ESEM evaluation of the theorized model at the scale-level

Initially, for comparison, the fit of the theorized four-factor oblique model was computed using CFA. The fit values were $\chi^2$ = 174.18, $df$ = 39, $p <$ 0.0001, CFI = 0.663, RMSEA = 0.127 (90% confidence interval = 0.108 to 0.147). Therefore, both the RMSEA and CFI values indicated poor fit. The fit values for the ESEM model were $\chi^2$ = 46.83, $df$ = 17, $p <$ 0.0001, CFI = 0.926, RMSEA = 0.091 (90% confidence interval = 0.060 to 0.122). For this model, the

**Table 2. Factor loadings for the MPQ-BF items in the ESEM model with 11 primary trait scales.**

| Item # | Primary Trait Scales | | | | | | | | | | |
|---|---|---|---|---|---|---|---|---|---|---|---|
| | WB | SP | ACH | SC | SR | ALIE | AGG | CONT | HA | TRAD | ABSO |
| M1 | **.38** | .08 | **.36** | .14 | .02 | **-.35** | .18 | .08 | .03 | .16 | .16 |
| M26 | **.39** | .19 | .03 | -.03 | -.35 | -.08 | -.01 | -.02 | .06 | .18 | .41 |
| M38 | **.46** | -.04 | .09 | -.05 | -.16 | -.27 | -.05 | -.12 | .06 | .16 | *.08* |
| M50 | **.59** | .06 | -.09 | -.03 | -.11 | -.04 | -.03 | .06 | .03 | -.08 | .01 |
| M62 | **.76** | -.07 | -.06 | -.14 | -.15 | -.04 | .19 | .10 | .08 | .00 | .03 |
| M74 | **.63** | .01 | .11 | -.03 | -.46 | .00 | -.11 | -.02 | -.16 | .04 | .06 |
| M85 | **.68** | .08 | -.01 | -.06 | -.05 | -.32 | -.20 | .03 | -.03 | .15 | .15 |
| M97 | **.51** | .11 | .00 | -.03 | -.03 | -.09 | -.04 | -.09 | -.01 | .16 | .28 |
| M109 | **.71** | .26 | .03 | -.02 | -.02 | -.04 | -.04 | -.21 | -.08 | .09 | .16 |
| M121 | **.59** | -.10 | .20 | -.04 | -.08 | -.28 | .04 | .18 | -.04 | -.12 | .18 |
| M133 | **.56** | -.01 | .20 | -.24 | -.13 | .00 | -.13 | -.19 | **.34** | .12 | -.06 |
| M144 | **.62** | .02 | .23 | -.19 | -.05 | -.31 | .00 | -.16 | -.02 | .09 | .15 |
| M2 | -.07 | **.86** | .06 | -.13 | .03 | .04 | -.18 | -.26 | -.11 | -.01 | .07 |
| M15 | **.40** | .35 | -.02 | -.26 | .24 | -.16 | **.32** | -.16 | .08 | -.14 | -.03 |
| M39 | -.07 | **.52** | .17 | .17 | -.25 | .07 | .27 | -.06 | -.17 | .12 | -.15 |
| M51 | -.13 | **.72** | .09 | .21 | .08 | -.07 | .09 | .25 | .12 | .05 | -.02 |
| M63 | -.09 | **-.31** | -.05 | .09 | .13 | .29 | -.04 | .08 | **-.38** | .15 | .12 |
| M75 | .07 | **.93** | .02 | -.02 | -.05 | .11 | -.05 | -.12 | -.05 | .06 | -.03 |
| M87 | .08 | **.64** | .12 | .10 | -.26 | .16 | .01 | .12 | .01 | -.19 | .11 |
| M98 | -.19 | **-.40** | .11 | .27 | -.15 | .29 | -.24 | .06 | -.15 | .24 | -.09 |
| M110 | .07 | **.96** | -.01 | .00 | .09 | .12 | -.13 | -.04 | -.03 | .05 | -.12 |
| M122 | -.13 | **-.33** | .08 | .20 | .26 | **.39** | -.23 | -.08 | -.13 | .16 | .01 |
| M134 | .13 | **-.41** | -.07 | .21 | -.17 | .15 | -.18 | -.20 | .23 | .03 | -.12 |
| M145 | .16 | **-.64** | -.01 | .09 | .14 | .16 | .04 | .05 | .15 | .03 | -.21 |

CFI value indicated adequate fit, and the RMSEA value indicated marginal fit. Therefore, the fit findings for this model was interpreted as having some acceptable fit.

Table 1 shows the factor loadings for the ESEM model. As demonstrated in the table, three of the four Positive Emotionality lower-order primary scales loaded positively and saliently (>.32) on the broad higher-order Positive Emotionality factor; the Absorption scale loaded positively and saliently on the Absorption factor; all three lower-order primary scales for Negative Emotionality loaded positively and saliently on the broad higher-order Negative Emotionality factor; and all three lower-order primary scales for Constraint scales loaded positively and saliently on the broad high-order Constraint factor. There was no salient secondary cross-loading (>.32 on a non-target factor). Although the lower-order Social Closeness factor only had a non-salient loading on it target Positive Emotionality higher-order broad factor, its loading on this factor was higher than its loadings on the other broad higher-order factors. Therefore, the four broad higher-order factors were considered to be well defined. Given this, and the adequate fir for this model at the global level, this model was interpreted as being adequate.

## Evaluation of the relationships of the factors in the EESM MPQ-BF model at the broad dimensions level with the factors in the BIS/BAS scales

Table 3 shows the correlations of the higher-order dimensions in the ESEM MPQ-BF model with the factors/scales in the BIS/BAS scales. As demonstrated, Positive Emotionality correlated positively with BAS-Drive. The magnitude of this correlation was of medium effect size,

**Table 3. Correlations of the ESEM MPQ-BF factors with the BIS/BAS scale scores.**

| MPQ-BF Factors | BIS/BAS Scale Scores | | | |
|---|---|---|---|---|
| | BIS | BAS-RR | BAS-DR | BAS-FS |
| Positive emotionality | -0.02 | 0.05 | 0.41*** | 0.07 |
| Absorption | 0.22 | -0.05 | -0.05 | 0.28** |
| Negative emotionality | 0.17 | -0.08 | -0.02 | 0.03 |
| Constraint | 0.39** | 0.13 | 0.05 | -0.64*** |

\***p < .001;

\**p < .01;

\*p < .05.

Absorption correlated positively with BAS-Drive, and the magnitude of this correlation was of small effect size. Negative Emotionality was not correlated significantly with any of the BIS/BAS factor scales. Constraint was correlated positively with BIS, and negativity with BAS-Fun Seeking. The magnitude of the correlation with BAS-Fun Seeking was of medium effect size, and the magnitude of the correlation with BIS was of large effect size.

## Discussion

The primary goal of the current empirical research was to use the ESEM procedure to assess the proposed dimensional structure of the MPQ-BF at the item-level (items loading on the 11 lower-order primary trait factors) and the scale-level (the 11 primary lower-order trait factors loading on the four higher-order broad dimensions) in a normative group of adults from the general community. At the item-level, although the RMSEA value indicated good fit, the CFI value indicated poor fit. Additionally, for this model, the factors were not clearly defined, with the primary lower-order trait factors not having all their designated items loading saliently on them, and having many non-designated items loading saliently on them. Therefore, overall, this model was interpreted as not demonstrating adequate support at the item level. In contrast to the findings here, the study by Eigenhuis et al. [6] that used the ESEM approach for examining the factor structure of the MPQ-BF at the item level found good global fit for the theorized model. However, because that study did not report the factor loadings for this solution, it is not known if the factors in the model were well-defined. As the present study examined the loadings and cross-loadings of the factor model at the item level, the conclusion reached is that this model is not an acceptable model for the MPQ-BF at the item level.

### MPQ factor structure

At the scale-level, the findings showed some degree of acceptable fit for the ESEM model. For this model, all but one lower-order primary trait factor loaded saliently on their targeted higher-order broad factors. The exception was Social Closeness. Although this scale had a non-salient loading on it target Positive Emotionality broad factor, its loading on this factor was higher than its loadings on the other broad factors. The findings also showed no salient cross-loadings on the higher-order broad factors. Therefore, all four factors of the ESEM MPQ-BF model were reasonably well defined, thereby supporting the four-factor oblique ESEM model at the scale-level. Previous PCA/EFA studies involving the MPQ-BF at the scales level have shown that although the primary factor scales loaded saliently on their target higher-order broad factors, there was considerable cross-loadings on the higher-order broad factors [4,8], thereby indicating that the higher-order broad factors in the MPQ-BF were not clearly defined. A likely explanation for the differences in the present study and that of previous studies is that unlike previous studies that used PCA and EFA

approaches, the present study used the ESEM approach [13]. The ESEM approach is generally considered a more realistic and superior approach than the PCA, EFA approach, and even the CFA approach [12,14]. Given this, it can be argued that the findings in the present study are more credible than that reported in past PCA/EFA studies. As far as it can be ascertained, this is the first study to examine the factor structure of the MPQ-BF at the scale level using ESEM. Although Eigenhuis et al. [6] used this approach, it only examined the factor structure of the MPQ-BF at the item-level. Additionally, because the ESEM approach was used, the global fit of our models were able to be examined. As already noted, the ESEM model showed adequate fit.

The support for the ESEM model at the scale-level was further enhanced in terms of support for the external validity of the factors in this model. Consistent with theory [6,17,18] and past findings [8], the findings here showed that Positive Emotionality correlated positively with BAS-Drive. Also, Constraint correlated positively with BIS, and negativity with BAS-Fun Seeking, with the correlation for BIS being stronger. Therefore, as suggested by Tellegen [18], the findings appear to suggest that the comparable constructs in the MPQ-BF for the BAS and BIS constructs in the initial version of RST are Positive Emotionality and Constraint, respectively. Another finding worthy of note is that Absorption correlated positively with BAS-Drive. Overall, taken together, these findings can be interpreted as providing reasonable support for the external validity of the factors in the ESEM model.

## Implications, limitations & further research

The findings in the study have implications for the use and revisions of the MPQ-BF. The findings in the study have implications for the use of the MPQ-BF. As the findings showed inadequate support for the factor structure of the MPQ-BF at the item level, it follows that the primary traits in this measure are not well defined by the proposed targeted items. Thus the targeted items cannot be reliably used to measure the primary traits, and if used to do so, the findings need to be viewed cautiously. However as there was adequate support for the four-factor structure at the scale level, it follows that the targeted scales do adequately measure the proposed four higher order factors in the MPQ-BF, and that they can be used to measure these traits. In this respect, it is important to note that the four-factor higher order model referred to here comprise the same factors proposed for the MPQ-BF in terms of how the 11 primary scales map on to the higher order dimensions. Given these considerations and our findings, it can be argued that only the MPQ-BF components comprising the factor structure involving the scales as indicators of the higher order factors be used for research and clinical practice. Given our findings that items did not clearly define their proposed primary traits, we believe that our findings, as such, do not have implication for reconceptualization of Tallegen's personality model.

Notwithstanding all these, our finding do underscore the need for revisions in the items in the MPQ. Table 2 shows the items with clean loadings (salient and no cross-loadings on other non-target factors) and problem loadings (non-salient and/or cross-loadings on other non-target factors) on their respective primary trait factors: It can speculated that if scores for the primary lower-order trait factors are desired it would be preferable to obtain scores from the sets of items with clean loadings rather the complete set of the 12 items. What this also means is that the revisions of the remaining items in the 11 lower-order primary traits scales (those listed under "" in Table 2 may be needed in future revised visions of the MPQ-BF. We stress though that the proposed set of possible item in a future revised version of the MPQ-BF is highly speculative, and that the reliability and validity of the ensuring MPQ-BF structured needs to be comprehensively examined in future studies before it is used.

In summary, the findings in the present study were interpreted as indicating that at the primary factor level, there was support for the ESEM model with four broad high-order factors (i.e., Positive emotionality, Absorption, Negative Emotionality, and Constraint). This model showed strong support in terms of external validity. In contrast, the proposed theoretical model for MPQ-BF at the item level was not supported. Although the present study provided valuable new psychometric findings for the MPQ-BF using an advanced methodology, the findings and interpretations in the study need to be considered in relation to several limitations. First, it is possible that factors such as age, gender, and ethnicity influenced ratings of MPQ-BF items. The failure to control for these effects in the study could have confounded the results. Second, because the study involved adults from the general community, it could be argued that the findings are unique to community samples, and cannot be generalized to clinical samples or special groups. Third, although the original intention was to obtain a random sample of adults, the eventual participants constituted a convenient sample. This may limit the generalizability of the findings and the conclusions made. Fourth, the external validity of the factors in the ESEM model were examined using a limited number of external variables, thereby limiting a more comprehensive evaluation of the external validity of the ESEM model. Fifth, is the relatively small sample size (N = ) in the study. Although a general rule of thumb is that a sample size of $\geq$ 200 is adequate for testing the theoretical CFA model [25], the simulation study by Bandalos [26], involving WLSMV suggests that at least 500 cases may be needed for sufficient power to reject models. It would be useful for future studies to examine the factor structure, external validity and other psychometric properties (such as measurement invariance) of the MPQ-BF at both the scale and item levels, taking into consideration the methodology used in the present study and the limitations highlighted.

## Supporting information

**S1 Data.**
(SAV)

**S2 Data.**
(SAV)

## Author Contributions

**Conceptualization:** Rapson Gomez, Mark D. Griffiths, Vasileios Stavropoulos.

**Data curation:** Rapson Gomez.

**Formal analysis:** Rapson Gomez.

**Methodology:** Mark D. Griffiths, Vasileios Stavropoulos.

**Software:** Rapson Gomez.

**Supervision:** Rapson Gomez.

**Validation:** Rapson Gomez, Vasileios Stavropoulos.

**Visualization:** Rapson Gomez.

**Writing – original draft:** Rapson Gomez, Vasileios Stavropoulos.

**Writing – review & editing:** Rapson Gomez, Mark D. Griffiths, Vasileios Stavropoulos.

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
