## [Decision Letter · Decision Letter 0]

2 Jan 2020

PONE-D-19-32430

Internal and External Validity of the Brief Version of the Multidimensional Personality Questionnaire: Exploratory Structural Equation Modelling

PLOS ONE

Dear Dr. Stavropoulos,

Thank you for submitting your manuscript to PLOS ONE. After careful consideration, we feel that it has merit but does not fully meet PLOS ONE’s publication criteria as it currently stands. Therefore, we invite you to submit a revised version of the manuscript that addresses the points raised during the review process.

As you will see below, both reviewers are favourable toward your manuscript and recommend only minor changes. If you are happy to make these changes, I would be happy to accept your manuscript without re-review. 

We would appreciate receiving your revised manuscript by Feb 16 2020 11:59PM. To enhance the reproducibility of your results, we recommend that if applicable you deposit your laboratory protocols in protocols.io, where a protocol can be assigned its own identifier (DOI) such that it can be cited independently in the future. For instructions see: http://journals.plos.org/plosone/s/submission-guidelines#loc-laboratory-protocols

We look forward to receiving your revised manuscript.

Kind regards,

Amanda N. Stephens

Academic Editor

PLOS ONE

Journal Requirements:

Reviewers' comments:

Reviewer's Responses to Questions

**Comments to the Author**

1. Is the manuscript technically sound, and do the data support the conclusions?

Reviewer #1: Yes

Reviewer #2: Yes

2. Has the statistical analysis been performed appropriately and rigorously? 

Reviewer #1: Yes

Reviewer #2: Yes

3. Have the authors made all data underlying the findings in their manuscript fully available?

Reviewer #1: Yes

Reviewer #2: No

4. Is the manuscript presented in an intelligible fashion and written in standard English?

Reviewer #1: Yes

Reviewer #2: Yes

5. Review Comments to the Author

Reviewer #1: The paper used ESEM models to examine the factor structure of the brief version of the Multidimensional Personality Questionnaire (MPQ-BR) both at the scale- and at the item-level. Results revealed poor fit and poorly defined factors at the item-level but supported the four-factor higher-order structure. Correlations with the BIS/BAS scales were in line with expectations.

Overall, the paper is well written and analyses sound correct. Results support the conclusions. However, some remarks should be pointed out.

Even though the results presented in the paper are interesting it is not stressed enough which may be their practical implications. The Authors suggested using a “reduced version” of the 11 primary lower-order trait scales (p 21): Would be reliable these scales? Which may be the results of using the reduced scales in terms of construct validity? Which may be the meaning of a non-defined structure at the item-level? How could be overcome these problems of the scale? Is it a problem of the scale or a matter of the underlying theory?

In addition, I would suggest to the Authors to consider shortening the paper by simplifying some paragraphs (e.g., first paragraph p 21, p 16). To this purpose, some tables would be useful.

Please, control the t-test (p 12).

Reviewer #2: Thank you for allowing me to contribute to the review of the manuscript “Internal and External Validity of the Brief Version of the Multidimensional Personality Questionnaire: Exploratory Structural Equation Modelling”.

The main strength of the present study involves the use of the novel and robust methodology of exploratory structural equation modelling (ESEM) to assess the structure of the Multidimensional Personality Questionnaire (MPQ). After careful evaluation of the manuscript I would like to note that the findings are somewhat controversial as they reveal acceptable fit at the scale level and poor fit at the item level.

It was refreshing to see that Tellegen's personality theory was adequately explained in the introduction of the manuscript, showing that a good literature review has been conducted. Nevertheless, the preference to this theory, or its advantages compared to other main personality theories and assessment methods are not entirely and clearly explained. In order to strengthen the manuscript, I would suggest the authors to add a paragraph addressing this issue to highlight the utility of the MPQ in the introduction.

The structure of the manuscript was carefully prepared and organised and the authors have employed third (advanced) generation statistical analyses to address their study aims. With this in mind, I believe the chosen analytical strategy was adequate to achieve the study’s aims.

The main weakness of this work involves the relatively small although normative sample recruited by the authors. I would suggest that the authors highlight more clearly this fact as a limitation in their limitation section.

Considering the implications, although briefly highlighted, I would suggest that the authors include a specific part about how the present findings discussing how they could inform Tellegen's theory in regards to specific potential conceptual modifications-reconsiderations.

Overall, I the present study provides a novel and valuable contribution to the field (considering in particular the psychometric examination of instruments with the novice ESEM methodology and conceptual considerations in Tellegen's theory). Thus, I would recommend accepting the present study provided minor revisions are conducted.

6. PLOS authors have the option to publish the peer review history of their article (what does this mean?). If published, this will include your full peer review and any attached files.

Reviewer #1: No

Reviewer #2: No

---

## [Author Response · Author response to Decision Letter 0]

24 Jan 2020

Response to Review Comments to the Author

Reviewer #1: 

Even though the results presented in the paper are interesting it is not stressed enough which may be their practical implications. The Authors suggested using a “reduced version” of the 11 primary lower-order trait scales (p 21): Would be reliable these scales? Which may be the results of using the reduced scales in terms of construct validity? Which may be the meaning of a non-defined structure at the item-level? How could be overcome these problems of the scale? Is it a problem of the scale or a matter of the underlying theory?

Response. Discussed in page 21 , para 1.

In addition, I would suggest to the Authors to consider shortening the paper by simplifying some paragraphs (e.g., first paragraph p 21, p 16). To this purpose, some tables would be useful.

Response. The section referred to has been shortened, with a new table (Table 3) to identify the relevant items (p. 21 para 2)

Please, control the t-test (p 12).

Response. No t-test findings was reported in the paper.

Reviewer #2: 

Nevertheless, the preference to this theory, or its advantages compared to other main personality theories and assessment methods are not entirely and clearly explained. In order to strengthen the manuscript, I would suggest the authors to add a paragraph addressing this issue to highlight the utility of the MPQ in the introduction.

Respone. This is provided in the revised paper (p. 7, para 2).

The main weakness of this work involves the relatively small although normative sample recruited by the authors. I would suggest that the authors highlight more clearly this fact as a limitation in their limitation section.

Response. This is cover in p. 22, para 2 (fifth limitation).

Considering the implications, although briefly highlighted, I would suggest that the authors include a specific part about how the present findings discussing how they could inform Tellegen's theory in regards to specific potential conceptual modifications-reconsiderations.

Response. Covered in p. 21, para 1.

---

## [Editor Report · Decision Letter 1]

7 Feb 2020

Internal and External Validity of the Brief Version of the Multidimensional Personality Questionnaire: Exploratory Structural Equation Modelling

PONE-D-19-32430R1

Dear Dr. Stavropoulos,

We are pleased to inform you that your manuscript has been judged scientifically suitable for publication and will be formally accepted for publication once it complies with all outstanding technical requirements.

With kind regards,

Amanda N. Stephens

Academic Editor

PLOS ONE
---

## [Editor Report · Acceptance letter]

14 Feb 2020

PONE-D-19-32430R1 

Internal and External Validity of the Brief Version of the Multidimensional Personality Questionnaire: Exploratory Structural Equation Modelling 

Dear Dr. Stavropoulos:

I am pleased to inform you that your manuscript has been deemed suitable for publication in PLOS ONE. Congratulations! Your manuscript is now with our production department. 

With kind regards,

on behalf of

Dr. Amanda N. Stephens 

Academic Editor

PLOS ONE